# New Therapeutics for Extracellular Vesicles: Delivering CRISPR for Cancer Treatment

**DOI:** 10.3390/ijms232415758

**Published:** 2022-12-12

**Authors:** Biying Yan, Yaxuan Liang

**Affiliations:** Center for Biological Science and Technology, Advanced Institute of Natural Sciences, Beijing Normal University, Zhuhai 519087, China

**Keywords:** extracellular vesicles, CRISPR, cancer, exosome, gene editing, gene therapy, Cas9

## Abstract

Cancers are defined by genetic defects, which underlines the prospect of using gene therapy in patient care. During the past decade, CRISPR technology has rapidly evolved into a powerful gene editing tool with high fidelity and precision. However, one of the impediments slowing down the clinical translation of CRISPR-based gene therapy concerns the lack of ideal delivery vectors. Extracellular vesicles (EVs) are nano-sized membrane sacs naturally released from nearly all types of cells. Although EVs are secreted for bio-information conveyance among cells or tissues, they have been recognized as superior vectors for drug or gene delivery. Recently, emerging evidence has spotlighted EVs in CRISPR delivery towards cancer treatment. In this review, we briefly introduce the biology and function of the CRISPR system and follow this with a summary of current delivery methods for CRISPR applications. We emphasize the recent progress in EV-mediated CRISPR editing for various cancer types and target genes. The reported strategies for constructing EV-CRISPR vectors, as well as their limitations, are discussed in detail. The review aims to throw light on the clinical potential of engineered EVs and encourage the expansion of our available toolkit to defeat cancer.

## 1. Introduction

Cancer has been the second leading cause of death worldwide, with the average age of incidence going down each year [1]. The major treatments for patients with cancer include surgery, chemotherapy, and radiotherapy, yet the recurrence rate, chemoresistance, and other side-effects have not been well addressed. Immunotherapies of growing interest are being actively evaluated in the real world [2,3,4,5,6]. Notably, chimeric antigen receptor T-cell (CAR-T) therapy against a range of leukemias and lymphomas has been approved by the FDA and made clinically available to patients [7]. Nonetheless, the principle of CAR-T therapy makes the application more suitable for liquid tumors. In the last decade, the development of gene editing tools has provided a new solution for cancer therapy. As is known, the occurrence of cancer is initiated by mutations or epigenetic changes in proto-oncogenes, tumor suppressor genes, and DNA repair genes [8,9,10]. The abnormal expression of such genes causes the dysregulation of subsequent signaling pathways and prompts cancer progression [11]. Gene editing technology aims to correct or disrupt the relevant mutational genes to reverse the disease. Notably, the clustered regularly interspaced short palindromic repeats (CRISPR)/CRISPR associated protein 9 (Cas9) system is well recognized as precise machinery for gene editing in cancer studies [12]. In spite of its great potential, CRISPR/Cas9-based technology has encountered major challenges in terms of delivery strategy and off-target effects. Nonviral and viral vectors have been intensively studied for the delivery of CRISPR components into mammalian cells ex vivo or in vivo, while limitations with respect to cell-type targeting specificity and delivery efficiency and safety concerns remain [13]. Extracellular vesicles (EVs) are nano-scale membrane-bound vesicles secreted by all known cell types [14]. EVs are packaged from parent cells with a variety of proteins, nucleic acids, lipids, and metabolites, which are essential to the functions of cell recognition, endocytosis, and alteration of recipient cell status [15,16]. Owing to the advantages in terms of cellular uptake, as well as the manipulative tactics for cargo recruitment, EVs are becoming ideal vectors for drug delivery and gene therapy. In this review, we introduce the basic biology of the CRISPR/Cas9 system and focus on the established methods for its delivery in vitro and in vivo. In particular, we highlight the emerging application of EV-mediated CRISPR/Cas9 gene editing in cancer research. Techniques utilized in the loading of CRISPR/Cas9 components into EVs are elaborately discussed in the order of plasmid DNA, mRNA, and proteins. Finally, we consider the challenges for EV-mediated gene editing and its clinical potential in cancer treatment.

## 2. CRISPR Function and Cellular Delivery

The CRISPR system was originally discovered in prokaryotes, in which it provides adaptive immunity against exogenous DNA invasion [12]. With deliberate molecular engineering, the system has been exploited as a powerful DNA editing tool for various cell types. The Cas endonuclease is guided by a non-coding RNA transcript to double-cut DNA strands at the intended site, followed by DNA ligation in a directed or random manner. However, delivering CRISPR/Cas components to mammalian cells in vitro or in vivo remains challenging. A plethora of strategies have been proposed to address delivery performance, including non-viral approaches and recombinant viral infection, though all have their own limitations.

### 2.1. Origin and Biological Functions of CRISPR/Cas9

CRISPR is a family of DNA sequences integrated into the genomes of a range of bacteria and archaea. As an immunological memory storage system, the sequences direct the synthesis of various non-coding RNA transcripts, and, together with Cas protein, build immunoprotection for most prokaryotic organisms [17]. A CRISPR locus consists of repeated sequences (30–40 bp in length) with variable numbers, which are separated by ~20 bp-sized spacers (Figure 1A) [18,19]. Spacers are derived from previously infected bacteriophages, and their variety and numbers are constantly being updated along with new viral attacks [17,18,20]. To acquire CRISPR-mediated immunity, foreign genome DNA is firstly digested into fragments and then inserted with the assistance of the Cas1-Cas2 complex as a spacer in between the CRISPR repeated regions [18,21]. When a new invasion is taking place, the CRISPR system immediately responds to direct the synthesis of RNA transcripts known as pre-crRNAs, which are comprised of spacer sequences and partially repeated regions (Figure 1B). Meanwhile, trans-activating CRISPR RNA (tracrRNA) is transcribed from a distinct locus that is also comprised of partially repeated sequences (Figure 1C). The pre-crRNA hybridizes with the tracrRNA transcript through the complementary base pairs within the repeated regions (Figure 1D). The binding with tracrRNA is essential for pre-crRNA maturation, in which the ribonuclease III (RNase III) engages and trims the duplex for the crRNA hybrid formation (Figure 1E) [22,23]. Next, the spacer region in the crRNA hybrid identifies the target and recruits Cas endonuclease to double-cut the foreign DNA strands, resulting in a disruption to DNA invasion (Figure 1F) [22,23,24]. In addition to crRNA targeting, a further requirement for the Cas/RNA complex’s cleaving of the DNA is the presence of a short conserved sequence named the protospacer adjacent motif (PAM) on the DNA to be severed [25]. The PAM is generally located downstream of the crRNA-binding region and serves as a binding site to the PAM-interacting (PI) domain of Cas9 [26]. The PAM interaction triggers Cas9 catalytic activity for DNA unwinding [27,28], and then the HNH domain and the RuvC domain on Cas9 respectively cleave the (+) strand and the (−) strand, where the (+) strand refers to the one complementary to the crRNA (Figure 1F) [25,29]. Specifically, the cut site occurs three or four nucleotides upstream of the PAM [25]. Of note, specific Cas variants recognize different sequences of PAMs. For instance, *S. pyogenes* Cas9 (spCas9) binds the “NGG” motif, while *Staphylococcus aureus* Cas9 (saCas9) tends to bind the “NNGRRT” motif [30,31]. There are several types of CRISPR/Cas systems found in nature with distinctions in RNA structures and Cas protein variations. Class 1, which includes types I, III, and IV, is characterized by a multi-subunit Cas complex for DNA cleavage. In contrast, the class 2 system, including types II, V, and VI, has an endonuclease module with a single-protein effector, which is more feasible for engineering [21,32,33]. Specifically, Cas9, as a single-chain endonuclease in the type II system, has been extensively studied to expand the genome toolkit [25,34].

When the CRISPR technique was introduced in mammalian cells, a single chimeric RNA that integrated crRNA and tracrRNA was developed to streamline guide RNA expression. In the design of single-guide RNA (sgRNA), a four-nucleotide linker “GAAA” called a tetraloop was commonly used to bridge the crRNA and the tracrRNA [27,30]. To direct Cas9 towards the intended cut site, the spacer region on the crRNA would be replaced by the ~20 nt sequence upstream of the potential cut point (Figure 1G). In addition, the determination of the cutting zone should concern the presence of PAM downstream of the cut site, due to the essential role that PAM plays in the activation of Cas9 [30]. Moreover, Cas9 entry into the nucleus is a requirement when the system is working in eukaryotic cells. It was shown that the SV40 nuclear localization signal (NLS) fused to the C-terminus of Cas9 improved the nuclear targeting of the gRNA/Cas9 complex [35,36]. In the nucleus, the Cas9/RNA complex captures the target DNA, unwinds the strands, and makes a doubt cut. Noticeably, in eukaryotic cells the DNA repair mechanism will be engaged soon after the cleavage occurs. The non-homologous end joining (NHEJ) pathway recruits multiple effectors in a short time to ligate the broken DNA, yet this repair has a great possibility of introducing random nucleotide insertions or deletions (indels), in which case a frame shift or a nonsense mutation is likely to happen, leading to a malfunction in the target gene [30,37]. The other DNA repair mechanism, the homology-directed repair (HDR) pathway requires a single-stranded DNA template to assist in the process of ligation; thus, by following the template sequence, DNA repair has much higher precision and fidelity [30]. In order to achieve a controlled DNA editing, efforts need to be made to promote the HDR pathway and suppress the more active NHEJ pathway. Studies have shown that a compound, Scr7, inhibited the DNA ligase IV, which is a key enzyme in the NHEJ pathway, and enhanced HDR activity by 19 folds [37]. Overall, by taking advantages of molecular engineering, the CRISPR/Cas9 system has developed as a substantial tool for precise DNA manipulation.

### 2.2. Established Methods for CRISPR/Cas9 Delivery

One challenge in utilizing the CRISPR/Cas9 toolset is the delivery of all essential effectors into a cell. The delivery may be implemented in the form of plasmid DNA that contains the Cas9 coding sequence, a sgRNA sequence, and/or a repair template, or instead the fragment of RNA that is ready for Cas9 translation once it has entered the cytosol. Other than nucleic acids, purified recombinant Cas9 proteins may be directly transferred into a cell and function immediately. For different purposes, the delivery approach involves the determination of a way to cross the cellular membrane, the time frame for endonucleolytic persistence, the cytotoxic effects of the vector, and the biodistribution when administrated in vivo. Herein, we briefly describe several established approaches for CRISPR/Cas9 delivery, including microinjection, which is used in individual cells, the electroporation of which depends on passive diffusion through temporary holes, lipid-based carriers designed for the in vivo purpose, and viral vectors with therapeutic potential. For the exploration of more CRISPR delivery strategies, readers are referred to excellent reviews elsewhere [38,39,40,41].

#### 2.2.1. Physical and Nonviral Approaches

Microinjection is a canonical way to send CRISPR/Cas9 components directly into the cytoplasm or nucleus. The technique is commonly used for single cells with large sizes, such as oocytes and zygotes. Petersen et al. injected a DNA plasmid containing CRISPR/Cas9 elements into swine zygotes to create an animal model with *GGTA1* biallelic knockout for the purpose of providing transplantable donor organs [42]. Similarly, Friedland et al. targeted the *nuc-119* or *dpy-13* genes by microinjecting CRISPR/Cas9 DNA into the gonads of adult *C. elegans* and finally established homozygous individuals by self-fertilization [43]. Except for germ cells, microinjection may not be the most suitable approach to deliver CRISPR/Cas9 because of the apparent damage to the cell membrane, the low productivity, and the requirement for operating experience. As a substitute, electroporation works on a mass of cells and enhances the delivery efficiency. With the pulse voltage and duration time thoroughly optimized, electroporation can be applied in both hard-to-transfect cells in vitro and living tissues. In a rodent model, Latella et al. injected CRISPR/Cas9 DNA into the subretinal space and finally managed to alter the retinal genome through repeated electric pulses given to the nearby tissue [44].

Lipid nanoparticles (LNPs) are delivery systems designed preferentially for in vivo purposes. Distinct forms of CRISPR/Cas9, including DNA, RNA, and ribonucleoprotein (RNP), are capable of being encapsulated in LNPs, though their encapsulation is largely dependent on the LNP formulation. Among the ingredients, the cationic ionizable lipid plays important roles in capturing and incorporating negatively charged nucleic acids or protein cargos. The stoichiometry of cationic lipids and their featured chemical groups further determine LNP cellular uptake and endosomal release. In addition, polyethylene glycol is frequently involved in formulations to prolong the particle circulation time in the body. LNP-mediated gene editing via the CRISPR/Cas9 system has been widely reported in the brain [45], lungs [46], liver [47], and tumor tissues [48]. Qiu and colleagues reported a newly identified hepatocyte-targeting LNP which carried Cas9 mRNA and gRNA and eventually knocked down the *Angptl3* gene in mice [49]. Another group achieved to deliver Cas9 RNP to the liver and lungs via intravenous injection of modified LNP contained 50–60% of the cationic lipid 1,2-dioleoyl-3-trimethypammonium-propane [50]. Overall, LNP carriers are beneficial in terms of their targeting specificity, biocompatibility, and good stability, while potential issues relating to toxicity and immune response still need pharmacologists’ attention.

#### 2.2.2. Viral Vectors

Packing the required elements of CRISPR/Cas9 into a viral vector provides possibilities to advance the gene editing towards the clinical setting. Currently, adeno-associated virus (AAV) vectors and the lentiviral vectors are broadly employed in gene delivery, and many of them are undergoing clinical trials [51,52]. Herein, we discuss these major vectors which were engineered to facilitate the delivery and expression of CRISPR/Cas9 essentials with a certain level of tissue tropism when administrated in vivo.

AAVs are members of the *parvovitidae* family and have a single-stranded DNA genome with a non-enveloped capsid shell for protection [53]. The genome of a wild-type AAV mainly consists of *rep* and *cap* genes responsible for AAV replication and packaging. In recombinant forms, these genes are removed and replaced with a promoter and the gene of interest to be expressed in the targeting cell. Due to the relatively small size (~20 nm in diameter) of AAVs, the maximum length of a DNA sequence that can be inserted into the AAV genome is ~4.7 k bases, which creates a problem when loading a spCas9 sequence with ~4.2 k bases plus other modules within a single virus. One way to solve the problem is to pack the DNA sequence for the sgRNA and the HDR template into a second AAV vector, though this approach doubles the workload of AAV production and the administrative titers [54,55]. AAVs have 12 or more serotypes, with their capsid proteins varying to a certain extent. Each serotype may have a distinct affinity to specific cell types, allowing AAVs to exhibit preferred tissue targeting features in in vivo treatments [56,57,58]. With further capsid manipulation, AAVs are capable of crossing the blood–brain barrier [59]. Thanks to their low cytotoxicity, pathogenicity, and immunogenicity, AAV-based gene therapies have been clinically tested in a range of disease treatments, including heart failure, neuromuscular disorders, severe hemophilia, and Parkinson’s disease, among others [51,60]. In a study of retinitis pigmentosa, CRISPR targeting of the *neural retina leucine zipper* (*Nrl*) gene was delivered via AAV8 to the subretinal tissue in a mouse model. Due to the capacity limit, spCas9 and sgRNA were individually packaged in separate AAVs. Finally, the AAV-CRISPR disrupted the *Nrl* gene and rescued the retinal degeneration [61]. In addition, Ekman et al. established a single AAV vector carrying saCas9 and sgRNA to break the mutant *huntingtin* gene in an R6/2 transgenic mouse model with Huntington’s disease. The AAV-mediated gene editing achieved an ~50% reduction in intranuclear inclusion in neurons, and the survival of the mice was prolonged [62]. However, it should be noted that most recombinant AAVs do not integrate their DNA cargos into the host genome. Instead, the delivered DNA is likely to freely co-exist in the nucleus. This feature provides an ideal long-term expression of delivered genes in a non-dividing cell, while it may also pose a challenge when treating actively dividing cancer cells.

Lentiviruses are a group of retroviruses which are comprised of an RNA genome, a capsid, and a glycosylated envelope. Lentiviruses infect both dividing and non-dividing cells and are able to insert transgenes into host genomes for stable and long-term expression [63]. To assemble a lentivirus for CRISPR/Cas9 delivery, a three-plasmid system is widely used, where psPAX2 is deployed for viral packaging, pMD2.G for VSV-G envelope expression, and lentiCRISPRv2 for Cas9 and sgRNA loading [64,65,66,67]. In contrast to small-sized AAVs, lentiviruses measure 80–100 nm in diameter, which allows a genome size of up to 9.7 k bases. The sufficient space offers the vector a great capacity to hold the Cas9 gene, sgRNAs, templates, and a further reporter gene. Some groups have generated lentiviruses carrying four sgRNAs, with each including an independent Pol III promoter. The vector worked to modify the target gene concurrently at four different sites [68]. Lentivirus-mediated gene editing has been adopted to generate stable cell lines and transgenic animal models; however, when projected to disease treatment, obvious risks arise due to the random genome integration which must be taken into consideration. Actually, the viral genome insertion may occur anywhere in the host DNA, disrupting normal genes and causing severe adverse events [69]. Further, lentivirus administration gives rise to the adaptive immune response and antibody neutralization, undermining follow-up delivery efficacy [70].

## 3. Extracellular Vesicle Biology and the Therapeutic Potential

EVs are heterogeneous membrane-enclosed nanoparticles derived from the cellular endomembrane system. EVs are produced by all known cell types, ranging from prokaryotes to eukaryotes, though the secreting mechanisms are thought to be completely distinct from one to another. In recent years, accumulating evidence has supported EVs as therapeutic vectors, with their functional components coming either from the parent cell’s endogenous sorting or from cell engineering for directed cargo loading. To date, EVs of various origins have been examined to treat cancers, the details of which are discussed below.

### 3.1. Extracellular Vesicles Biology and Subclassification

In multicellular organisms, EV biogenesis and secretion is considered a significant mechanism for intercellular communication [71]. EVs are initiated from cellular endosomes or the plasma membrane, during which various molecular cargos, including soluble and membrane proteins, nucleic acids, lipids, and metabolites, are sorted within the vesicles. Released EVs disperse in the proximal extracellular matrix or participate in the circulation for the targeting of distal cells [72,73]. Many studies have revealed that EV targeting and entry is an active and energy-consuming process. Vesicle uptake starts with ligand–receptor interactions on the cell surface, followed by direct fusion with the plasma membrane or a more complex internalization towards the endosomal–lysosomal pathway [71]. Of note, the cellular uptake of EVs may be fulfilled via distinct routes, including micropinocytosis, macropinocytosis, clathrin-dependent endocytosis, clathrin-independent endocytosis, and phagocytosis, the exact mechanisms being not fully understood [74,75]. Nevertheless, once taken up by the recipient cell, the enrichment of EV cargo molecules would occur and eventually alter the physiology of the cell [74].

As mentioned above, EV biogenesis has two major routes. The endosome-mediated pathway requires multiple steps originally from the invagination of plasma membrane, which forms the early endosome in the cytosol (Figure 2A). The early endosome later matures to the late endosome with a more regular and round morphology. The membrane microdomains on the late endosome further bud inward to form many intraluminal vesicles (ILVs), which are the precursors of EVs [71,76]. The late endosome with ILVs inside (also known as the multivesicular body) is directed to fuse with the plasma membrane and dump EVs outside the extracellular environment [74,77]. Alternatively, some multivesicular bodies have been observed to finally be involved in lysosome fusion and protein recycling, the mechanism of which, however, has not been fully unveiled (Figure 2A) [78]. EVs produced from this endosome-related route can be named as exosomes. The second route EV forms is through the outward budding directly from the plasma membrane, the subpopulation of which may be named as microvesicles or ectosomes (Figure 2B). These EVs’ generation is also regulated and depends on specific microdomains on the plasma membrane, and the cargo profile or molecular enrichment may vary markedly from parent cells and their exosome counterparts [79,80]. 

Even though the EV biogenesis routes are relatively clearly depicted, huge challenges remain in the EV subtype classification, which is mainly due to the limited availability of EV purification methods. Currently, many EV isolation techniques exist including ultracentrifugation, size-exclusion chromatography, ultrafiltration, polymer-based sedimentation, etc., however, none of them has the isolation principle related to the EV biogenesis. In this matter, unless the biogenesis pathway is demonstrated, the widely-seen “exosomes” or “microvesicles” may no longer be proper to use to name EV isolates. In fact, the International Society of Extracellular Vesicles has updated guidelines for EV study in 2018, and advocates a more stringent nomenclature for EV subpopulations [81]. Specifically, vesicles with sizes smaller than 100 or 200 nm can be named as small EV (sEV), which are regularly seen in the pellet of high speed centrifugation (e.g., >100,000× *g*). Vesicles with sizes over 200 nm are named as large EV (lEV), which may be collected from a lower speed spin (e.g., 10,000~20,000× *g*). Subgroups of EVs can further be classified by their density or protein biomarkers (e.g., CD81, CD63, CD9, Flotillin-1, Alix, etc.) according to the way of vesicle purification or characterization. Of note, the EV biogenesis does not dictate the size or biocomposition specificity of vesicles, which implies, for instance, that exosomes may or may not contain typical CD81/CD63 markers [82,83], or microvesicles have lEVs as well as sEVs below 200 nm in diameter [84,85,86]. Such heterogeneity of EV subpopulations and miscellaneous but technically-confined isolation methods render huge challenges to standardize the quality of EV study across laboratories in the whole community, and in this article we encourage a wider acknowledgement of guidelines proposed in the “minimal information for studies of extracellular vesicles 2018” (MISEV2018) by EV researchers [81].

### 3.2. EV Therapeutics for Cancer Treatment

EVs carry a number of cargo molecules, the functions of which are well evidenced as being capable of altering recipient cells. Based on the comprehension of vesicle composition, EVs have a noted potential to be developed as encouraging therapeutic vectors to treat diseases. Compared with other drug delivery carriers, such as liposomes, viral vectors, and nanoparticles, EVs have a more complex membrane composition that assists the vesicles in traveling across the tight junction barriers to achieve selective cellular uptake [87,88]. EV therapeutics for cancer treatment needs to consider the vesicle origination events that define EV cargo profiles and the key players. Leucocytes have attracted much attention in the pathology of cancer progression, and their interaction with cancer cells is thought to be crucial to cracking the cancer problem. Macrophages in the M1 activated form secrete multiple cytokines and promote the pro-inflammatory response [89]. EVs released from M1 macrophages were shown to activate naïve macrophages through stimulating the NF-κB pathway. The released cytokines (e.g., iNOS, IL-6, and IL2) from EV-activated macrophages consequently induced the apoptosis of breast cancer cells via the caspase-3 pathway [90]. M2 macrophages are generally thought to participate in immunosuppression and tissue regeneration, and their secreted EVs were found to support tumor growth [91,92]. Interestingly, when transfected M2 macrophages expressed downregulated lncRNA SBF2-AS1, which is a promotive effector for various cancer types, the released EVs incorporated increased levels of miR-122-5p and showed inhibitory effects towards pancreatic cancer [92]. Natural-killer (NK)-cell-derived EVs were also evidenced to inhibit tumor growth; exosomal miR-186 was confirmed to prevent immune escape in neuroblast patients [93]. In addition, NK-cell EVs were manipulated to recruit let-7a mimics or siRNAs against BCL-2 to further enhance the anti-cancer effects [94,95]. Similarly, dendritic-cell-derived EVs were able to activate T cells, B cells, and NK cells through the delivery of tumor antigens, which is regarded as a potential alternative to tumor vaccination [96,97].

In addition, cell engineering is another way to develop EV-based cancer therapies. By introducing recombinant proteins or overexpressed nucleic acids to parent cells, the biomolecules of interest may be increasingly sorted towards EVs and play essential roles in recipient cancer cells. Among those utilized for EV production, HEK293 and CHO cells have been considered ideal candidates due to their high transfectivity and low safety concerns [98,99]. Remarkably, an engineered platform that employed suspension HEK293 cell lines was established and the antisense oligonucleotides that overexpressed EVs underwent clinical trials against advanced hepatocellular carcinoma, colorectal cancer, and non-small cell lung cancer [100]. Mature red blood cells (RBCs) lack a range of organelles and in culture release abundant EVs upon stimulation [101]. Evidence suggested that RBC EVs had more resistance to freeze–thaw cycles without severe aggregation and were free of concerns related to genomic or mitochondrial DNA contamination [102]. In some proof-of-concept studies, RBC EVs presented favored uptake by breast cancer cells or acute myeloid leukemia cell lines, the EV-loaded antisense oligonucleotides sufficiently suppressing cancer cell proliferation [101].

## 4. EV-Mediated CRISPR/Cas9 Delivery for Cancer Therapy

Cancer is a genetic disease that accumulates DNA alterations pivotal to cell growth and division; thus, as a genetic tool, CRISPR/Cas9 is thought of as a promising therapeutic approach to cure cancer. However, it remains challenging to deliver CRISPR/Cas9 components in vivo in a safe and efficient manner. Given the advantages in terms of biocompatibility, structural stability, limited immunogenicity, and low cytotoxicity, EVs are ideal candidates for the packaging and delivery of the CRISPR/Cas9 system for cancer treatment. Recently, emerging studies have focused on EV-encapsulated CRISPR/Cas9 as a novel strategy to edit the gene profile in the host cell (Table 1). In this section, we summarize reported applications of EV-CRISPR treatment in cancer studies, and then discuss major engineering approaches to fabricate EV-CRISPR/Cas9 vectors.

### 4.1. EV-CRISPR Administration Targeting Various Types of Cancers

Many labs have evaluated the outcomes of EV-CRISPR/Cas9 delivery and DNA cleavage in cell lines. Ran et al. produced a stable HEK293T cell line with the EGFP gene inserted into the genome. Upon treatment with CRISPR/Cas9-containing EVs, EGFP expression was significantly reduced, suggesting the function of EV-based DNA editing [114]. Another lab generated a reporting system in the lung carcinoma cell line A549, where a stop element was added in between the promoter and the DsRed coding sequence. EV-CRISPR/Cas9 was designed to target the stop element, and, after the treatment, the stop element was undermined and the A549 cells started to glow [112]. These earlier studies tested the feasibility of EV/CRISPR function in mammalian and cancer cells. To further advance EV-based treatment for cancer, a couple of oncogenes have attracted interest from researchers. Zhuang et al. examined the interruption of WNT10B, an overexpressed oncogene involved in the Wnt family, in a hepatocellular carcinoma cell line, HepG2, using an EV-CRISPR/Cas9 vector. The intravenous administration in mice reduced the size of tumor explants in a dose-dependent manner [109]. As another oncogene target, microRNA-125b (miR-125b) overexpression was commonly found in patients with acute myeloid leukemia (AML). Further studies revealed that miR-125b promoted the proliferation of leukemic cells and inhibited their differentiation [115,116]. Usman et al. utilized cargo-manipulated EVs derived from red blood cells to deliver Cas9 mRNA and sgRNA into an MOLM-13 cell line and observed a marked decline in miR-125b expression. However, the group did not report phenotypic change in leukemic cells after DNA editing, although they showed reduced cell proliferation via EV-mediated delivery of miR-125b antisense oligonucleotides [101]. *myc* family genes are associated with plenty of cancer types, including Burkitt lymphomas (BLs), where the *c-myc* gene is dysregulated in B lymphocytes due to chromosomal translocation [117]. A study showed that EV-delivered Cas9 endonuclease sufficiently disrupted MYC expression and decreased cell proliferation and viability by activating cell apoptosis; however, an in vivo trial to determine EV-CRISPR/Cas9 efficacy was not included in the study [103]. *KRAS ^G12D^* is a Wnt-pathway-associated mutation found in the initiation and maintenance of pancreatic cancer [118,119]. By employing EVs as vehicles for CRISPR/Cas9 delivery, McAndrews et al. demonstrated a slowdown of KPC689 cell growth in vitro, as well as suppression of implanted pancreatic tumor tissues, in a mouse model [106].

Beyond directly targeting oncogenes, other strategies that combine gene editing with chemotherapy have been developed to treat cancers. Poly (ADP-ribose) polymerase-1 (PARP-1) is a nuclear enzyme which actively responds to DNA damage and is engaged in the DNA repair pathway [120]. Inhibition of PARP enhanced the sensitivity of cancer cells to DNA-damaging agents (e.g., alkylators, platinum, etc.), giving rise to an improvement in cancer treatment [120,121]. In a study on ovarian cancer, EVs containing both the CRISPR/Cas9 system and cisplatin compounds were fabricated and assessed in an SKOV3 cell line and a murine model bearing tumor tissues. The EV-delivered Cas9 complex disrupted PAPR-1 expression both in vitro and in vivo, which synergistically enhanced the performance of cisplatin in killing the cell [105]. Similarly, EV-mediated delivery of a kinase inhibitor, sorafenib, in combination with CRISPR/Cas9 elements targeting IQ-domain GTPase-activating protein 1 (IQGAP1), an overexpressed oncogene widely reported in hepatocellular carcinoma, was shown to augment drug lethality with respect to apoptosis induction and growth suppression [104]. Overall, EV-CRISPR/Cas9, as a novel anti-cancer vector, has drawn much interest from plenty of laboratories and its implementation has been demonstrated in a broad spectrum of cancer types and target genes.

### 4.2. Tactics Used to Fabricate EV-CRISPR Vectors

Given the ~100 nm diameter for EVs, the intravesicle space provides sufficient room to load varying numbers and kinds of CRISPR/Cas9 components. EV-delivered DNA plasmids do not integrate into the genome of the recipient cell, yet they still produce relatively long-term endonucleolysis. EV-mediated RNA has a shorter half-life but a more prepared expression. EV-delivered Cas9 protein functions immediately upon uptake by the recipient cell, though tactics should be considered to bypass lysosomal degradation. For different forms of cargo, researchers have proposed suitable methods for EV recruitment. Electroporation may be a reasonable option to incorporate CRISPR cargos because EVs have lipid membranes similar to cells. Additionally, EV biogenesis includes many cargo-sorting machineries, of which advantage can be taken to manipulate the recruitment of Cas9 mRNA, protein, and RNP in situ. Here, we review details of methods reported for packaging CRISPR/Cas9 elements into EVs and examine their existing limitations from the perspective of their clinical potential.

#### 4.2.1. Plasmid DNA Encapsulation for EV-CRISPR/Cas9

EVs are able to deliver CRISPR/Cas9 in the form of a plasmid vector that contains sequences of Cas9 and/or sgRNAs lined up in a certain way. However, the DNA-sorting machinery is still controversial in EVs [122,123]. Though some studies have reported that microvesicles may naturally incorporate plasmid vectors from parent cells [124] while others have suggested that cancer-derived EVs contain genomic DNA fragments [125,126,127], a verified method that promotes DNA sorting within EVs has not been established. Electroporation with short high-voltage pulses has been an applicable approach that allows plasmid diffusion through temporary pores on the EV membrane [103,104,105]. He et al. generated a Cas9 protein stably expressing cell line and observed Cas9 recruitment in EVs. The purified EVs were then loaded with plasmids containing sgRNA elements by electroporation and consequently showed a capacity for genome cutting [104]. Nonetheless, drawbacks of electroporation have been noted in view of EV morphology preservation after pulses and the sophistication required in optimizing voltage parameters [128]. Further, the passive diffusion towards EVs was largely dependent on the size of the plasmid, and any construct over 4000 base pairs was not able to enter the EVs [129]. To avoid those problems, an EV–liposome hybrid method was proposed, in which purified EVs were incubated with plasmid DNA-associated liposomes for 12 h at 37 °C, finally generating vesicles containing plasmids in the lumen [107]. It should be noted that liposomes had DNA attached to the exterior, yet how the DNA crossed the membrane during the fusion was not explained. The hybrid vesicle with the CRISPR/Cas9 plasmid was taken up by mesenchymal stem cells and downregulated *Runx2* gene expression [107]. Beyond the packaging technique, it is worth considering other issues when using CRISPR/Cas9 DNA for gene editing. EV-released plasmids in recipient cells require regulated transportation through nuclear pores; however, the molecular mechanism underneath remains unclear [130,131]. Further, the long-term existence of foreign plasmids in host cells may cause overwhelming activity of genome cleavage, which undermines the purpose of gene correction [132,133]. Overall, plasmid DNA employed for EV-CRISPR/Cas9 is straightforward, yet it is subject to loading uncertainties and uncontrolled risks.

#### 4.2.2. Cas9 mRNA Sorting for EV-CRISPR/Cas9

EVs are well evidenced to encapsulate and functionally transfer RNA to recipient cells [134,135]. Usman et al. developed a method to load transcripts embracing the Cas9 coding region and the gRNA sequence to RBC-derived EVs through electroporation. The modified EVs sent RNAs to a leukemia cell line, MOLM13, and suppressed the expression of miR-125a and miR-125b two days post-treatment [101]. Interestingly, the authors evaluated the loading performance via electroporation and revealed that the large-sized plasmid was much less well incorporated than Cas9 mRNA, evidencing the advantage of RNA as opposed to plasmid DNA loading for EV-mediated CRISPR/Cas9 editing [101]. As discussed above, the electric approach causes extra damage to EV structure and integrity. Efforts to develop strategies to sort and package RNA during the process of EV biogenesis are of great importance. EVs are known to conserve a set of recruiting mechanisms for mRNA, microRNA, long noncoding RNA, circular RNA, and beyond [136]. Though the RNA packaging mechanism remains largely unknown, a few sorting signals have been uncovered [137,138]. The RNA-binding protein HNRNPA2B1 was responsible for EV enrichment of long RNA transcripts that shared specific sequence characteristics [139,140]. Another sequence of 12 nucleotides was confirmed as an mRNA sorting signal to EVs in hepatic cells [141]. In addition, a few RNA motifs along with their binding proteins were uncovered in the EV miRNA-sorting mechanism [142,143,144,145]. Inspired by such RNA loading schemes, Li et al. generated a recombinant protein that fused the cytosolic end of CD9 with human antigen R (HuR), an RNA-binding protein recognizing AU-rich elements (AREs) [108,146]. When the Cas9 transcripts were overexpressed in the cell, the HuR module captured the mRNA through interaction with the 3x AREs domain located downstream of the Cas9 open reading frame. With the assistance of CD9 for EV targeting, the recombinant complex with the Cas9 mRNA was packaged into the vesicle [108]. The authors tested the function of CD9-HuR EVs for the purpose of targeting C/ebpa, and the in vivo experiments demonstrated the inhibition of the gene in the liver [108]. Taken together, harnessing the endogenous EV RNA-sorting pathway for CRISPR/Cas9 RNA encapsulation streamlined the production workload and preserved vesicle integrity. The delivered RNAs in the recipient cell were ready for translation without bothering to enter the nucleus, giving a more rapid response to CRISPR/Cas9 functioning. In addition, the half-life of imported RNAs is manageable without risks of genome insertion or long-term existence. This approach has to some extent been limited by the lack of understanding of EV RNA-sorting mechanisms, yet it will benefit from our accumulated comprehension of EV biogenesis and cargo selection.

#### 4.2.3. Cas9 Protein or RNP Acquirement for EV-CRISPR/Cas9

Considering the fact that the synthesis and purification of recombinant Cas9 proteins is well established, a more straightforward way to build an EV-CRISPR/Cas9 system is to directly load the protein or the Cas9/sgRNA RNP into a vesicle. Similarly, physical approaches, such as puncturing vesicle membranes to allow molecular diffusion towards EVs, have been reported. Zhuang et al. tested sonication and freeze–thaw cycling individually to permeabilize EVs for Cas9/sgRNA RNP entry [109]. Distinct from the high-voltage pulse used in electroporation, sonication disrupts lipid membrane rigidity and microviscosity through ultrasound waves, while the freeze–thaw cycles produce ice crystals in the vesicle that stab the membrane and produce holes [109,147,148]. Though a ruptured membrane would subsequently recover to a certain extent at 37 °C incubation, EV remodeling due to vesicle fusion or aggregation likely alters the characteristics of EV trafficking and uptake [109,149,150]. To avoid the problem, another lab attempted to apply a protein transfection reagent (e.g., PULSin^®^) to EVs for Cas9 loading [110]. The synthetic lipid nanoparticles from the reagent bear positive charges, which, once Cas9 proteins are attached, interact with negatively charged EV membranes and finally send the cargo proteins into the vesicles. In the study, the authors observed improved cellular uptake and limited cytotoxicity from EV-PULSin-Cas9 compared to PULSin-Cas9, which oftentimes impacted the viability of treated cells due to the cationic profile of the nanoparticles [110]. The EVs utilized in the study were released from human MDA-MB-231 breast cancer cells; however, evidence suggested that cancer-cell-derived EVs may contain RNA species that promote the proliferation of recipient cells, which may compromise the conclusion regarding the cytotoxicity of EV-PULSin-Cas9 [110,151].

Beyond incorporating proteins into isolated EVs, sophisticated strategies for sorting Cas9 proteins to the vesicle throughout EV biogenesis have been designed. Interestingly, evidence indicated that Cas9 RNP could be packed into EVs when parent cells were stably transfected with CRISPR/Cas9 components by lentiviruses, although the recruitment efficiency may not be sufficient [111]. To enrich the loading, EV targeting proteins were employed. A group generated a CD63 fusion protein with a C-terminal GFP which was able to maintain the EV-targeting capability [112]. Meanwhile, researchers fused Cas9 protein to a nanobody against GFP; this was a single-chain antibody fragment with a high affinity for GFP [112]. Attributed to the GFP–nanobody interaction, Cas9 was nearly doubly incorporated into the EVs as compared with the strategy that did not involve nanobody fusion and consequently altered reporter gene expression in the recipient lung cancer cells [112]. Similarly, Yao and co-workers attached Com protein to the N- or C-terminus of CD63 for EV enhancement of Cas9 RNP [113]. The Com protein was a translational regulator derived from bacteriophages and bound to the secondary structure of a specific RNA sequence known as *com* [152]. When the tetraloop region on an sgRNA was replaced by *com*, the Cas9/sgRNA complex was shown to be captured by the CD63-Com fusion protein for EV targeting. The replacement of the sgRNA tetraloop was reported with an unrecognized impact on the functioning of Cas9/sgRNA RNP. The system proved to be applicable for both saCas9 and spCas9 and was able to integrate multiple distinct sgRNAs for concurrent gene editing [113]. Though the employment of CD63 advanced Cas9 RNP targeting to EVs, the system required co-transfection of multiple plasmid constructs, which increased the complexity of the EV production. Whitley et al. attempted to address the problem by adding a short peptide at the N-terminus of Cas9 for EV sorting. The peptide was derived from Src family kinases with a preference of myristoylation by N-myristoyl transferase 1. This modification added a long carbon chain to the glycine residue of the peptide, serving as a signal to the membrane localization. Intriguingly, 1.47% of total proteins in wild-type EVs were modified with the myristoyl group with unknown functions [114]. As expected, the myristoylated Cas9 was more likely than the native Cas9 to be sorted within EVs. However, the gene editing efficacy of EV-mediated myristoylated Cas9 was impaired, decreasing from 32.7% to 23.6%, probably because attachment of the lipid abolished the nuclear localization of Cas9 in the recipient cell [114]. In summary, the CRISPR presentation in EVs opting for protein or RNP was endorsed, yet the caution should be noted that the manipulation of Cas9 protein suffered the risk of impaired nuclear transportation and accurate genome localization.

## 5. Discussion and Clinical Prospects

CRISPR technology is an impressive achievement of the last decade. The rapid progress in the field is greatly moving the application forward to clinical settings. Since the first clinical trial of CRISPR-based gene therapy began in 2018 for the treatment of sickle cell disease and β-thalassemia, many more trials for genetic diseases, endocrine diseases, infectious diseases, and inflammatory diseases have been initiated and approved in the past few years by regulatory authorities in the United States, Europe, and other countries. Although the ethical discussion about CRISPR editing is still ongoing, it should be noted that all current clinical trials restrained the DNA changes in somatic cells or specific tissues without any inheritance opportunity arising. Among the current clinical trials, only a few have opted for in vivo delivery of CRISPR/Cas9. AAV or LNP vectors carrying CRISPR DNA have been injected directly into the eye or systematically via intravenous routes, respectively. As stated in an earlier section, the delivery of CRISPR DNA faces the great danger that lasting Cas endonucleolytic activity results in unexpected cuts in the genome. Hope may be found in CRISPR RNA- or RNP-based therapy; however, the viral vector is not the one properly doing the job. In the rest of the clinical trials that have been carried out, the strategy was combined with cell therapy, the CRISPR/Cas9 system being delivered ex vivo into harvested cells. After screening, the edited cells were infused back into patients intravenously. This strategy applies to all current trials for cancer where the CRISPR/Cas9 system has aided in the generation of engineered CAR-T cells targeting leukemia or lymphoma. Actually, the majority of CRISPR editing techniques for solid tumor cancers remain at the proof-of-concept stage, primarily owing to the lack of a proper toolbox for in vivo delivery.

An ideal delivery vector for the treatment of most carcinomas would have low off-target uptake, limited immunogenic toxicity, high drug-bearing capacity, and adequate penetration depth. EVs have opened up the possibility of developing next-generation therapeutic vehicles against cancers. EV surface engineering on membrane proteins or phospholipids reshapes the molecular profiles and possibly enhances vesicle targeting specificity. The major approaches to modifying EV uptake characteristics have been deliberately addressed elsewhere [153,154,155]. The adverse immunogenic reactions from either autologous or allogeneic EVs have rarely been reported; however, the complexity and heterogenicity of EV cargo molecules should prompt caution among pharmacologists. In fact, an entire list of EV ingredients has never been catalogued, not to mention the huge discrepancies emerging from varying donor cell types and statuses. This sets a challenge for EV quality control and standard establishment when heading for clinical translation. Nevertheless, EVs have been well received for their diversified loading strategies, as highlighted above. Remarkably, the possibility of encapsulating Cas9 RNA or RNP has potentially addressed the major drawbacks of viral or nonviral vectors, though a reliable method is lacking for measurement of the loading ratios of Cas9/sgRNA quantities and vesicle numbers. Our lab encourages the establishment of a repeatable and reliable methodology allowing the evaluation of EV-CRISPR/Cas9 loading efficiencies across laboratories. With the assistance of advanced vehicles, CRISPR treatment for solid tumors has more possibilities with respect to development. Considering the data from our and other labs showing that systematically infused EVs tend to stick in the lungs and liver, we foresee more opportunities for clinical translation in carcinomas in these tissues. In conclusion, EV-mediated CRISPR therapeutics represents a powerful toolset for a range of disease management strategies, and it is believed that it will become a beneficial asset in prospective precision medicine.

## Figures and Tables

**Figure 1 ijms-23-15758-f001:**
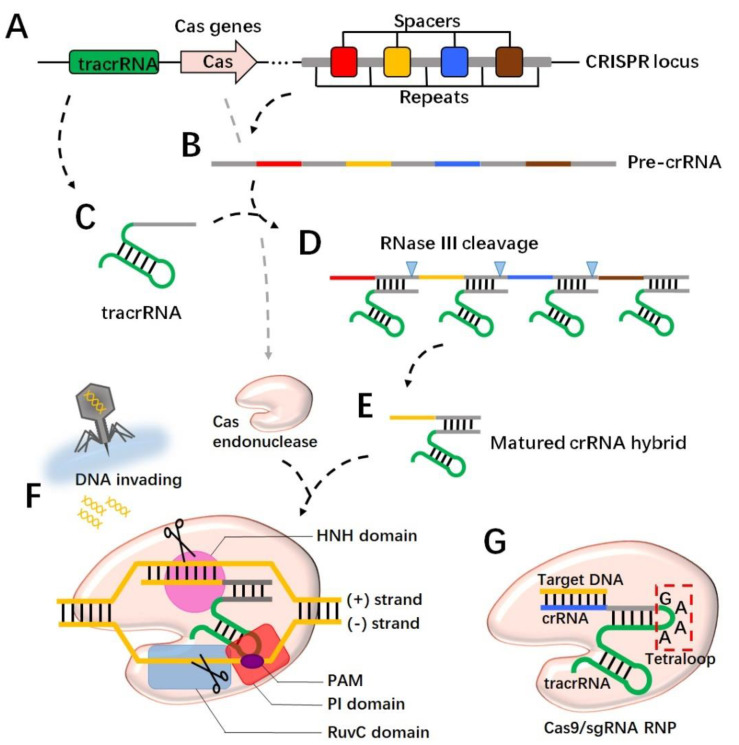
CRISPR cleaves DNA strands through RNA-guided Cas protein. The CRISPR locus in the prokaryotic genome has alternating spacers and repeats (**A**), which are transcribed for pre-crRNA (**B**). The tracrRNA is transcribed from a different locus in the genome (**C**) and binds to pre-crRNA through the complementary sequence to the repeat region (**D**). The binding with tracrRNA recruits RNase III, which trims pre-crRNA for matured crRNA (**E**). Cas endonuclease forms the ribonucleoprotein (RNP) with crRNA and targets the invading DNA through spacer sequence recognition (**F**). A PAM motif in the targeted DNA is required to activate Cas at the PI domain, and consequently the HNH domain and the RuvC domain cut the (+) and (−) strands of the DNA, respectively (**F**). In an engineered version, crRNA connects with tracrRNA to from single-guide RNA (sgRNA) by a 4 bp linker called a tetraloop (**G**). The spacer region is replaced to identify the target DNA. Together with Cas9, the RNP makes a double cut.

**Figure 2 ijms-23-15758-f002:**
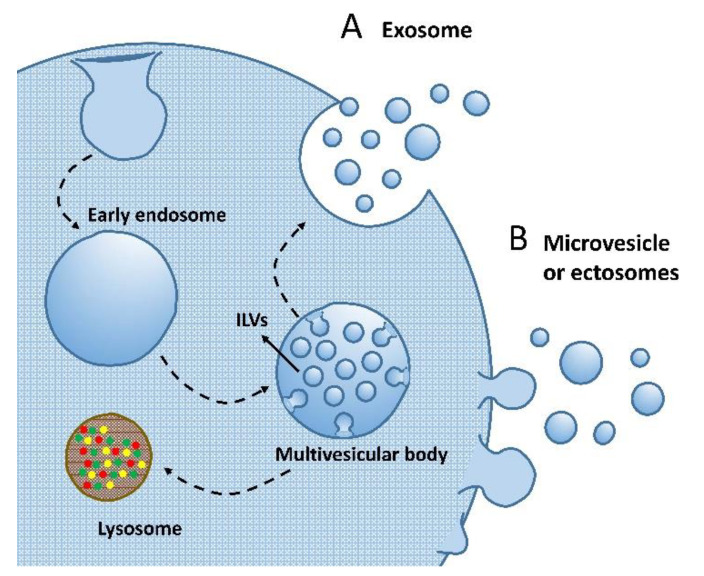
Routes of EV biogenesis. (**A**) Exosome biogenesis depends on the cellular endomembrane system, with the plasma membrane invaginating to form endosomes. Endosomes further mature to multivesicular bodies with intraluminal vesicles (ILVs) enclosed. The multivesicular bodies either go to lysosomes for degradation or fuse to the cell membrane for exosome release. (**B**) Microvesicles or ectosomes directly bud from the cell surface. Both generation machineries may give vesicles with a variable size distribution.

**Table 1 ijms-23-15758-t001:** Summary of EV encapsulation of CRISPR for gene editing in vitro and in vivo.

EV Source	CRISPR Form	EV Loading Method	Target Gene	Target Cell/Animal	Major Outcome	Ref.
HEK293T	Plasmid	Electroporation	MYC	Raji (B lymphocytes), Nalm6 (B cell precursor leukemia cells)	Induced cell apoptosis, inhibited tumor size	[103]
HEK293T	Plasmid	Electroporation	IQGAP1	HuH7 (liver cancer cells)	Induced cell apoptosis, in vitro only	[104]
SKOV3	Plasmid	Electroporation	PARP-1	SKOV3 (ovarian cancer cell), SKOV3-grafted mice	Inhibited cell proliferation, induced apoptosis, inhibited tumor growth	[105]
HEK293T	Plasmid	Exo-Fect™ Exosome Transfection Kit (SBI)	Kras^G12D^	KPC689 (pancreatic tumor cell), KPC689-grafted mice	Suppressed cell proliferation, attenuated tumor progression	[106]
HEK293T	Plasmid	EV/liposome fusion	Runx2, CTNNB1	Murine MSC	Suppressed protein expression	[107]
RBCs	Cas9 mRNA + sgRNA	Electroporation	miR-125b	MOLM13 (acute myeloid leukemia cells)	Suppressed miR-125b expression	[101]
HEK293T, AML12 (mouse hepatocytes)	Cas9 mRNA + sgRNA	CD9-HuR + Cas9 mRNA with ARE motif	C/ebpa	Adipogenic stem cells, mice	Suppressed C/ebpa expression	[108]
HEK293T	Cas9 RNP	Sonication, freeze–thaw cycling	WNT10B	HepG2, HepG2-grafted BALB/c nude mice	Reduced cell viability, inhibited tumor progression	[109]
MDA-MB-231	Cas9 RNP	EV/cationic lipid nanoparticle (PULSin^®^) fusion	Not examined	Raw 264.7 (macrophages), MDA-MB-231 (breast cancer cells)	Unaffected cell uptake efficiency, negligible cytotoxicity	[110]
HepAD38, HuH7, Vero, CHO, Hela	Cas9 RNP	Cas9 overexpression	HBV DNA, HPV DNA	HepAD38 (HBV-expressive liver cells), HuH7 (liver cancer cells), Hela (cervical cancer cells)	Cut HBV or HPV DNA in cells	[111]
HEK293T	Cas9 RNP	CD63-GFP + Cas9-nanobody	The stop element	A549^Stop-DsRed^ (lung cancer cells)	DsRed expression	[112]
HEK293T	Cas9 RNP	CD63-COM + sgRNA-*com*	DMD intron 50, intron 51, exon 53	DMD reporter cell, del52hDMD/mdx mice	Multiplex cleavage of target genes	[113]
HEK293T	Cas9 RNP	Cas9 myristoylation	EGFP	EGFP stably expressed HEK293T	Downregulated EGFP expression	[114]

Abbreviations: DMD, Duchenne muscular dystrophy; MSC, mesenchymal stem cell; RBC, red blood cell; RNP, ribonucleoprotein.

## Data Availability

Not applicable.

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
