# Peer review of "New Therapeutics for Extracellular Vesicles: Delivering CRISPR for Cancer Treatment"

_ijms, 2022, doi:10.3390/ijms232415758_

Round 1

Reviewer 1 Report

Dear Authors,

The manuscript is well-written and enjoyable to read. It nicely covers the published literature on the CRISPR/Cas9 delivery vehicle advantages and disadvantages related to cancer treatment.

To give a correct view of current cancer immunotherapies in the Introductions, it is suggested to include a reference to FDA approved CAR-T therapies (https://hillman.upmc.com/mario-lemieux-center/treatment/car-t-cell-therapy/fda-approved-therapies).

The Discussion on clinical prospects could be more elaborated to give a future perspective on which cancer types could be treated with CRISPR/Cas9 approach and in combination with other therapies or as a stand-alone treatment.

Reviewer 2 Report

Dear Editor

Dear Authors

The manuscript entitled “New Therapeutics for Extracellular Vesicle: Delivering CRISPR for Cancer Treatment” explains CRISPR system followed by a summary of current delivery methods for CRISPR application. This review paper also includes the recent progress in the EV-mediated CRISPR editing for varying cancer types and target genes.

The manuscript can be considered for publication in IJMS. However, the following points must be considered before further processing:

1- The title of the paper is misleading. It is suggested to revise and find a new title to demonstrate the major focus of the paper (recent progress in the EV-mediated CRISPR editing for varying cancer types). The first part of the title (New Therapeutics for Extracellular Vesicle) can be deleted.

2- Section 2.1. Origin and biological functions of CRISPR/Cas9:

This section is too long and has been discussed in several other papers with details. It is highly suggested to summarize this section and explain it by illustration (as shown in Figure 1).

3- Section 2.2. Methods for CRISPR/Cas9 system delivery:

Since this part is a fundamental section, it is necessary to be discussed more precisely. Please see section 10 in this paper (https://doi.org/10.1016/j.jconrel.2020.06.038) or this paper (https://doi.org/10.1016/j.addr.2021.113891) and explain the different methods for CRISPR/Cas9 system delivery.

4- 2.2.1 Physical and non-viral approaches:

This section does not explain any delivery system based on liposome or cationic polymers or peptides. It is suggested to add a table to summarize the most recent viral and non-viral delivery systems used for CRISPR delivery.

5- 3. Extracellular vesicle biology and the therapeutic potential:

Are Evs viral or non-viral delivery systems? If they are categorized as non-viral delivery system, then they must be discussed with the other non-viral delivery methods. In the current edition, it is assumed that EVs are a separate and independent delivery method rather than conventional viral and non-viral systems.

6- 3.2. EV therapeutics for cancer treatment:

This section is irrelevant to the other parts of the paper and can be deleted. If the authors want to keep it, the section must be re-written.

7- Table 1.

It is suggested to add a column to mention the major outcome of each study.

8- 4.1 EV-CRISPR administration targeting various types of cancers:

This section is a collection of some raw data without any discussion. This part must be revised or integrated into the other parts.

9- The conclusion section is poor and must be improved. For examples, the immunologic aspects, storage condition of delivery system and major bottleneck limiting bench to bedside translation of CRISPR technology can be discussed.   

Reviewer 3 Report

The review is well organized and well written. I further have no critical comment for this manuscript. 
